

# Effect of elevation, season and accelerated snowmelt on biogeochemical processes during isolated conifer needle litter decomposition

Laura T. Leonard[1], Eoin L. Brodie[2], Kenneth H. Williams[2,3] and Jonathan O. Sharp[1,4]

[1] Department of Civil and Environmental Engineering, Colorado School of Mines, Golden, Colorado, United States
[2] Lawrence Berkeley National Laboratory, Berkeley, California, United States
[3] Rocky Mountain Biological Laboratory, Crested Butte, Colorado, United States
[4] Hydrologic Science and Engineering Program, Colorado School of Mines, Golden, Colorado, United States

Corresponding author
Jonathan O. Sharp,
jsharp@mines.edu

## ABSTRACT

Increased drought and temperatures associated with climate change have implications for ecosystem stress with risk for enhanced carbon release in sensitive biomes. Litter decomposition is a key component of biogeochemical cycling in terrestrial ecosystems, but questions remain regarding the local response of decomposition processes to climate change. This is particularly complex in mountain ecosystems where the variable nature of the slope, aspect, soil type, and snowmelt dynamics play a role. Hence, the goal of this study was to determine the role of elevation, soil type, seasonal shifts in soil moisture, and snowmelt timing on litter decomposition processes. Experimental plots containing replicate deployments of harvested lodgepole and spruce needle litter alongside needle-free controls were established in open meadows at three elevations ranging from 2,800–3,500 m in Crested Butte, Colorado. Soil biogeochemistry variables including gas flux, porewater chemistry, and microbial ecology were monitored over three climatically variable years that shifted from high monsoon rains to drought. Results indicated that elevation and soil type influenced baseline soil biogeochemical indicators; however, needle mass loss and chemical composition were consistent across the 700 m elevation gradient. Rates of gas flux were analogously consistent across a 300 m elevation gradient. The additional variable of early snowmelt by 2–3 weeks had little impact on needle chemistry, microbial composition and gas flux; however, it did result in increased dissolved organic carbon in lodgepole porewater collections suggesting a potential for aqueous export. In contrast to elevation, needle presence and seasonal variability of soil moisture and temperature both played significant roles in soil carbon fluxes. During a pronounced period of lower moisture and higher temperatures, bacterial community diversity increased across elevation with new members supplanting more dominant taxa. Microbial ecological resilience was demonstrated with a return to pre-drought structure and abundance after snowmelt rewetting the following year. These results show similar decomposition processes across a 700 m elevation gradient and reveal the sensitivity but resilience of soil microbial ecology to low moisture conditions.

## INTRODUCTION

Coniferous forests are at high risk of ecosystem stress as climate disturbances amplify globally (*Seidl, Schelhaas & Lexer, 2011*). Drought-induced stress reduces tree canopy cover, net productivity, and rhizospheric processes (*Schlesinger et al., 2016*) with potentially lasting implications for terrestrial organic matter cycling and associated microbial activity (*Hyde et al., 2016*; *Negrón & Cain, 2018*). Hydrologic shifts associated with warming trends in snow-dominated mountain environments, such as shorter periods of snow cover, have further implications for downstream water availability (*Barnett, Adam & Lettenmaier, 2005*). Litter decomposition plays an important role in carbon and nitrogen cycling and can also be affected by drought conditions (*Santonja et al., 2015*). As a result, projected increases in global temperatures and shorter periods of snow cover could exacerbate poor soil health and ecology in mountain ecoregions (*De Vries et al., 2018*).

The North American Rocky Mountains largely consist of coniferous forests and the region has observed a variety of climate change impacts. Rocky Mountain National Park reported a 1.9 °C rise in average annual temperature in the last century with snow now melting 2–3 weeks earlier. Collectively these climate shifts have been associated with increased conifer tree mortality along with hydrologic implications for the Colorado River watershed (*NPS, 2018*). Moreover, a recent study identified biomes that harbored a heightened risk for organic carbon release from soils in association with climate change (*Kramer & Chadwick, 2018*). Within the Rocky Mountains, these biomes include polar wet tundra and boreal wet forests that encompass montane and subalpine ecoregions of Colorado (*GeoNetwork, 2007*). Classifying sensitivity broadly by biome alone is limiting however due to the heterogeneity of local mountain terrain (*Wainwright et al., 2020*). Specifically, sensitivity to drought varies across locations with slope and aspect influencing energy balances and soil moisture (*Körner, 2007*). Soil type also can impact soil moisture and geochemical signatures (*Lawrence & Hornberger, 2007*; *Wilson et al., 2008*) as well as the variability of snowpack dynamics across elevation that can play an additional role in drought responses across ecoregions (*Tennant et al., 2017*).

Soil microbial community responses to environmental stress additionally can influence decomposition and biogeochemical cycling processes. Within this study, we focus on different species of tree litter as the type of forest litterfall can affect associated microbial decomposition processes. Specifically, high quality litter typically defined by lower C:N ratios and higher bioavailability impact community composition by stimulating soil microorganisms and increasing decomposition rates (*Šantrůčková, Krištůfková & Vaněk, 2006*; *Fraterrigo, Ream & Knoepp, 2018*). Our past work determined that lodgepole litter was more bioavailable than spruce as evidenced by lower lignin-based aromatics and higher $CO_2$ respiration rates after soil rewetting (*Leonard et al., 2020a*). Beyond the litter decomposition processes, soil conditions are known to affect microbial communities with reduced activity associated with low moisture conditions and in turn increased $CO_2$ rates

after rewetting of these soils (*Birch, 1958*; *Schimel, 2018*). The cause of this effect as it relates to microbial mechanisms and whether resilience to drought drives these observations is not fully understood. Recent studies have shown microbial resilience to drought after rewetting cycles in a variety of environments (*De Nijs et al., 2019*; *Zhou et al., 2019*) in which increased species richness contributes to functional redundancy with implications for carbon storage (*Allison & Martiny, 2008*).

The goal of this study was to (1) determine the role of soil type, elevation, and early snowmelt on biogeochemical processes during litter decomposition, (2) evaluate the effects of litter presence, type, and seasonal shifts in soil moisture on microbial decomposition processes, and (3) understand the resilience of soil microbial ecology to low moisture conditions. Within our study domain, we hypothesized that soil respiration rates and microbial community richness would be greatest during summer rewetting events. In addition to these biogeochemical indicators, needle decomposition and organic carbon export would be most pronounced at the mid-elevation where moisture and temperature were less limiting. By extension, we expected enhanced needle decomposition and organic carbon export at the mid-elevation with early snowmelt further enhancing decomposition and microbial respiration rates.

To address these research questions, we isolated lodgepole and spruce needles from the hydrobiogeochemical complexity of the forest. Harvested needles were deployed in similar slope and aspect open meadows representative of three ecoregions within Colorado. This allowed insight into the specific roles of tree litter on local soil biogeochemistry. Measurements including gas flux, porewater chemistry, microbial community structure, and changes in needle mass and chemistry were monitored over three years (2017–2019) and across three elevations ranging from 2,800–3,500 m. Snowmelt was accelerated annually at the middle elevation adjacent to a control plot that underwent natural snowmelt rates to enable further investigation of snowmelt timing. The experiments encompassed large differences in snowpack and precipitation during a historically warm period, further enriching the observed impacts of climate change on local biogeochemical cycling (*Colorado State Forest Service, 2020*).

## MATERIALS & METHODS

### Needle collection and deployment

Conifer needles were collected from regions of Colorado identified in our prior work and transported to established field plots (*Brouillard et al., 2017*; *Leonard et al., 2020a*). This enabled us to study isolated decomposition processes in response to climate-associated variables in the field. Lodgepole pine (*Pinus contorta*) needle litter was harvested from the White River National Forest (39.54° N, 106.15° W, elevation 3,000 m) while Engelmann spruce (*Picea engelmannii*) needles were harvested from Monarch Pass (38.50° N, 106.33° W, elevation 3,400 m). The needles were harvested beneath respective tree stands in August of 2016 and air-dried at room temperature in a dark location. The dried needles were sieved with a Tyler Equivalent 10 mesh (1.7 mm) to separate debris and homogenize samples. Aliquots of 250 g for each needle type were weighed and separated for deployment in October 2016.
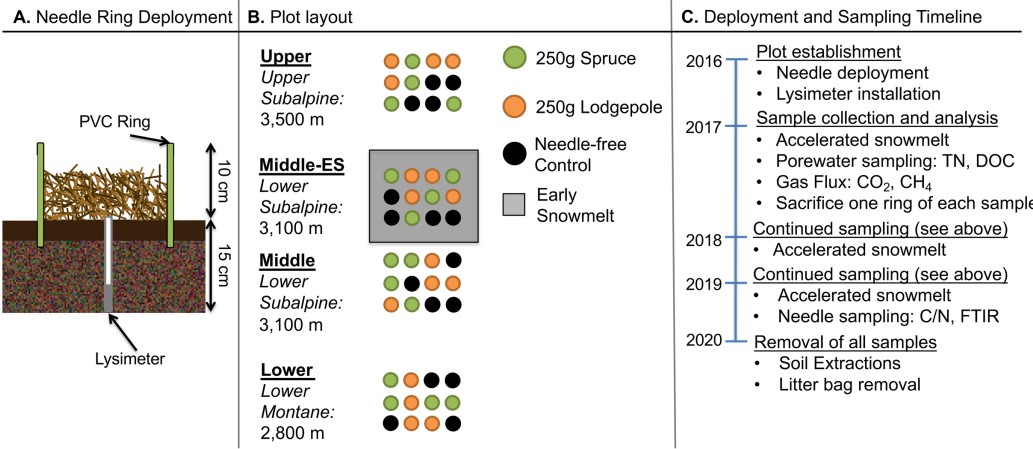

**Figure 1 Experimental design, plot, and schedule.** (A) The schematic of the ring deployment, (B) plot layout, and (C) sampling timeline are provided. Four replicate collars for each sample type were deployed in Fall 2016. In October of 2017, one of each sample quadruplicate was sacrificed to inform additional studies published elsewhere (*Leonard et al., 2020a*).

Experimental plots were established at three elevations. An additional plot at the middle elevation was subjected to accelerated snowmelt for a total of four distinct plots. The U.S. Department of Agriculture and Forest Service approved the field permit (Authorization ID: GUN1120). In order to focus on litter decomposition with minimal interference from flora, native foliage was raked to remove aboveground plant matter and belowground roots within the footprint of deployed 25 cm diameter and 18 cm height polyvinyl chloride (PVC) rings. The rings were pounded into the ground leaving approximately 10 cm of the ring aboveground. Lysimeters (Soil Moisture 1905L06 15 cm) were installed in the middle of each ring for future porewater collections (Fig. 1A). A total of 48 rings were deployed across the three elevations with a duplicate plot at the middle elevation consisting of four replicates of the spruce, lodgepole, and needle-free controls as depicted in Fig. 1B. The controls were bare soil within the rings while the spruce and lodgepole rings were filled with 250 g of the respective needles. Randomization was ensured using R Studio version 3.5.2 to produce 4 × 4 matrices with sample assignments as depicted in Fig. 1B. The experimental plots were monitored during the snow-free months of May through October during the years 2017, 2018 and 2019. In October of 2017, one of each sample quadruplicate was sacrificed to inform additional studies published in *Leonard et al. (2020a)*. This reduced sample replicates to three thereafter. The needle deployments limited native plant growth within the rings, however any observed emergent plant growth within the sample rings was manually pulled including the roots monthly to minimize native foliage interferences. Temporal soil biogeochemical analyses including soil extractions, gas flux, porewater and microbial DNA were conducted with sampling frequency and totals dependent on each method and media type sampled (Fig. 2).

In August of 2017, needle litter bags were deployed at each elevation to more effectively quantify mass loss associated with decomposition. The needle bags were constructed by wrapping 2 g each of archived lodgepole and spruce needles in a single layer of white nylon mesh fabric (23 × 23 openings per 2.54 cm) that was further wrapped in one layer of
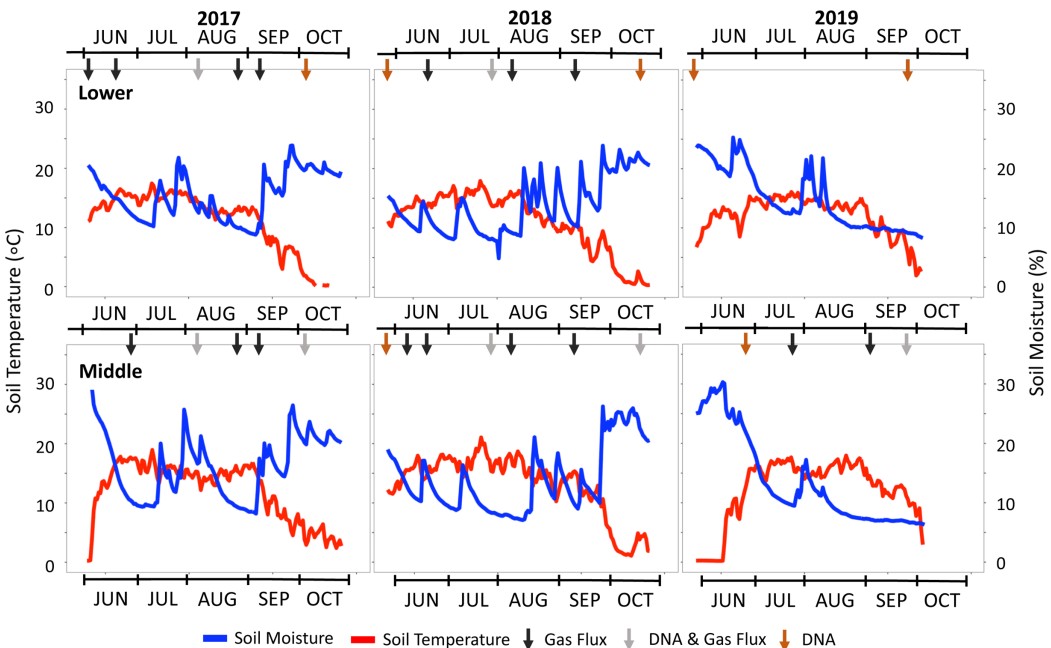

**Figure 2** **Seasonal soil temperature and moisture data across the three study years.** Climate data is shown for the three study years with soil moisture and temperature data from the Lower (2,800 m) and Middle (3,100 m) locations. The Upper plot data is not shown but is provided in Fig. S2. Arrows indicate specific sampling events conducted as indicated in black (gas flux), grey (DNA & gas flux), and orange (DNA).

window screen (17 × 13 openings per 2.54 cm). This contained the needles in a 5 × 5 cm flat surface area. Quadruplicates of the litter bags were placed level to the ground inside newly emplaced PVC rings. The PVC rings were deployed according to the same methods as the sample rings by removing native foliage and maintaining bare soil. Mass balance from each elevation after three years of needle decomposition (August 2017–August 2020) was calculated by the differences in weight from the initial 2 g and the final 2020 needles in each litter bag.

## Experimental plot descriptions

The experimental field plots were located in Crested Butte, Colorado within the Washington Gulch drainage of the East River watershed in designated areas approved by the Rocky Mountain Biological Laboratory. Field plots were established at three ecoregions: Lower Montane (38.92° N, 106.95° W, elevation 2,800 m), Lower Subalpine (38.96° N, 107.03° W, elevation 3,100 m), and Upper Subalpine (38.97° N, 107.04° W, 3,500 m) (Fig. 1B). These ecoregions are within the identified boreal wet forest and polar wet tundra Holdridge life zones (GeoNetwork, 2007). The study plots were established within northeast/ east facing aspect meadows with an average slope of 7–12° and will be referred to as the "Lower", "Middle", and "Upper" plots throughout the paper. The soil types of each location were unique with Tilton sandy loam at the Lower plot, a mixture of Tilton sandy loam, Cryaquolls and Histosols at the Middle, and Bucklon silt loam at the Upper plot (United States Department of Agriculture, 2019). The dominant bedrock

material of Crested Butte in the study location is Sedimentary Clastic Mancos Shale with a mixture of Middle Tertiary Intrusive Igneous (*U.S. Geological Survey, 2005*).

Snowmelt timing increased with elevation and the earliest snow-free dates occurred at the Lower plot (Table S1). Duplicate plots at the Middle elevation were established to further contrast the effects of snowmelt timing (Fig. 1B). Early snowmelt manipulations were implemented once isometric snowpack conditions were established in the middle of April (April 10–25) from 2017–2019. During deployment, a permeable 50% black mesh fabric was carefully secured on top of the snow over the duplicate Middle plot for approximately one month as described in *Leonard et al. (2020b)*. The tarp was removed shortly after the plot had fully melted and while snowpack within the control plot remained. These two Middle plots will be referred throughout the paper as the "Middle-ES" for the early snowmelt plot and "Middle" for the control snowmelt plot.

A snow telemetry (SNOTEL) station near the study locations on Mt. Crested Butte (38.89° N, 106.95° W, elevation 3,100 m) provided climate records for local weather data (Fig. S1). SNOTEL data for daily maximum air temperature, maximum soil moisture percentages at 20 cm depth, and snow water equivalent (SWE) precipitation were tabulated for each study water year and averaged to determine climate normals for comparison. Water years were determined by the twelve months starting on October 1st and ending on September 30th of the following year. Averaged 20-year normals were calculated from 1990–2010 for all climate variables except soil moisture, which only had reliable data sufficient for a 10-year average from 2006–2016. In addition to SNOTEL data, unshielded microclimate sensors buried at roughly 12 cm were established at each of the four experimental plots to monitor hourly soil moisture and temperature that were used to determine daily average values. Microclimate measurements were recorded from 2017–2019 at the Lower and Middle plots and 2017–2018 at the Upper plot (Fig. S2).

## Needle, soil, and porewater compositional analyses

A chemical analysis of the 2016 harvested and 2019 decomposed needles focused on temporal shifts in carbon, nitrogen, and organic functional groups. These analyses were conducted with small subsamples collected over a 3-year window of less than 5% of the total needles. The collected needles were air-dried, homogenized, and ground to a powder (*Leonard et al., 2020a*). Samples were submitted for total C and N measurements using a LECO TruSpec CN analyzer (LECO Corporation, St. Joseph, MI, USA). A subset was submitted for Fourier Transform Infrared (FTIR) analysis at Lawrence Berkeley National Laboratory. Per the methods of *Leonard et al. (2020a)*, the FTIR absorption spectra were baseline corrected in R Studio version 3.5.2 using R package hyperSpec (*Beleites & Valter, 2018*) and peak-by-peak integration analysis was conducted to quantify organic functional groups of interest. Specifically, biological-related functional groups (*Naumann, 2000*) observed in litter and bark studies (*Pandey & Pitman, 2003*; *Özgenç, Durmaz & Kuştaş, 2017*; *Yang et al., 2019*) were targeted for sample comparisons. The prominent peaks identified within this study were polysaccharides (900–1,200 cm$^{-1}$), cellulose-based ether linkages (1,157–1,159 cm$^{-1}$), lignin-based aromatics (1,508–1,510 cm$^{-1}$), amide I (1,600–1,700 cm$^{-1}$), and carbonyl typically at 1,738 cm$^{-1}$.

Soil extractions were conducted in 2020 in association with plot deconstruction. Soil was collected from the upper 2–3 cm soil horizon in contact with needle decomposition, homogenized on-site, and frozen at −20 °C until analysis. Deionized (DI) water extractions of soils followed the methods from *Leonard et al. (2020a)*. In brief, samples were sieved at 2 mm and a 2:9 solid mass to liquid weight ratio using 40 mL liquid volume for extraction in 50 mL centrifuge tubes. The tubes were placed on a shake plate for one hour, then centrifuged and filtered at 0.45 µm. Aliquots of the filtrate were acidified with hydrochloric acid and analyzed for total nitrogen (TN) and dissolved organic carbon (DOC) using a Shimadzu TOC-550A Total Organic Carbon Analyzer. The remaining filtrate was analyzed by ion chromatography for nitrate and nitrite, by colorimetric assays for ammonia, and by specific UV absorbance (SUVA) with a DU 800 Spectrophotometer. Extraction results were normalized to dried soil weights.

Porewater yields were sufficient during the shoulder months of spring and fall. All porewater collections were conducted on the same date when full snowmelt was established at each plot. Soil porewater was collected from the lysimeters in each sample ring over 12 h with 50mL Soil Moisture locking vacuum syringes to ensure sufficient water yields. The syringes were covered in canvas cloth to prevent sample interference from solar radiation. The porewater samples were filtered at 0.45 µm and frozen the same day of collection for storage at −20 °C until analysis. The porewater was analyzed for TN, DOC, and SUVA following the same methods for the filtrate from the soil DI water extractions. Porewater sampling was not possible at the Lower plot in 2018 due to dry conditions.

## Soil gas flux

Gas flux measurements were conducted in 2017–2019 at the Middle plots (14 sampling events) and during 2017–2018 at the Lower plot (nine sampling events) (Fig. 2). Equipment access was not feasible at the Upper plot. Gas flux measurements were conducted using a Picarro G2508 cavity ring-down spectroscopy analyzer capable of measuring $CO_2$, $CH_4$, $N_2O$ and $NH_3$. A closed system was established within the headspace of each sample ring by placing an airtight PVC chamber over the deployed rings. Norprene tubing was used to circulate cavity headspace from the PVC chamber to the analyzer inlet and outlets. Flux data collection typically occurred midday. Sample collection began when steady state was determined by linear trends of production or removal, initiating a two-minute data collection period for each sample. Fluxes were calculated using the Hutchingson & Mosier method provided in the Picarro software using the calculated headspace volume. The sampling sequence was randomized during each measuring event to minimize temporal biases associated with hourly fluctuations in temperature. Further, the two plots at the Middle elevation were measured simultaneously during measurements, alternating between the Middle and Middle-ES plots for each sample.

## Soil microbial DNA extraction & rRNA gene sequencing

Soil for DNA extractions was collected from surficial soils (<2–3 cm depth) after gently pushing aside existing needles. Within the area of the sample ring, no less than 10 grab samples were collected with autoclaved spatulas to fill a two mL sample collection tube up

to 1 g for each sample ring. The grab samples were randomized throughout the ring's surface area to minimize biases. DNA extraction and amplification were conducted for each sample in singlet following the methods of *Leonard et al. (2020a)*. In brief, soil DNA was extracted using the ZymoBIOMICS DNA Miniprep kit and 0.25 g of soil were weighed out from the 2 mL sample collection tube after homogenizing the total sample mass. The 5 PRIME Hot MasterMix and a primer set using the 515-Y forward and 926R reverse primers were used to amplify the 16S and 18S ribosomal RNA genes (*Parada, Needham & Fuhrman, 2016*). Sample barcoding with the amplicons was completed using an adapted forward primer with the M13 sequence (*Caporaso et al., 2012*; *Stamps et al., 2016*). Final DNA concentrations were determined using the Qubit 2.0 fluorometer. Libraries were sequenced at the Duke Center for Genomic and Computational Biology using an Illumina MiSeq instrument and V2 PE250 chemistry. The raw sequencing reads were processed as described in *Leonard et al. (2020a)* using the DADA2 package (*Callahan et al., 2016*) and taxonomic assignments with Silva v128 (*Pruesse, Peplies & Glöckner, 2012*). After quality filtering the 16S genes a total of 2,500,000 sequences were obtained with a sequence depth range of $9,765 \pm 4,270$. Normalization was performed by rarefaction at 4,299 which retained 224 out of 254 samples. Similarly, 102,000 sequences were obtained after filtering the 18S genes with a sequence depth of $400 \pm 420$. Due to the high variance in depth, rarefaction was performed at 200, retaining 143 out of 254 samples. Raw sequence data have been deposited in NCBI SRA databases under access number PRJNA 605259 for the Middle site 2017–2018 samples previously published in *Leonard et al. (2020a)* and PRJNA 715914 with the rest of the samples. Rarified samples were further processed in R Studio version 3.5.2 using the Phyloseq (v1.26.1) R package for data sorting, analysis, and visualization (*McMurdie & Holmes, 2013*) in addition to statistical analyses and visualization packages described in the next section.

## Statistical analyses

Due to low sample sizes ($n = 3$), Friedman and *post hoc* Wilcoxon tests were not used in this study for statistical determinations. As a result, mean and associated standard deviation values for environmental triplicates were used as alternatives when appropriate for geochemical parameters including needle composition, porewater, soil extractions and gas flux. Possible significance between samples was indicated by non-overlapping error bars. Gas flux data were binned by year to assess linear Pearson correlations between annual flux and soil parameters.

DNA sequencing data were analyzed for beta and alpha diversity. Beta diversity of samples binned by elevation (Lower, Middle, and Upper) was determined using the Adonis function as an analogous multivariate analysis of variance using weighted and unweighted UniFrac distance matrices in the community ecology R package vegan (*Oksanen et al., 2019*). The Adonis test was followed by principal coordinate analyses (PCoA) performed on the weighted and unweighted UniFrac distances. In addition, the dual Middle plots (Middle and Middle-ES) were compared separately from the full dataset to determine if significant differences were present as a function of snowmelt. Next, the same methods of the Adonis function followed by PCoA were conducted across all samples as a function

**Table 1 Climate variable shifts across each study water year.**

| Water year (Oct–Sep) | Dry soil[*] | Hot days[**] | Total precipitation (cm SWE) | Snow (cm SWE) |
|---|---|---|---|---|
| 2017 | 143 | 73 | 94 | 61 |
| 2018 | 163 | 88 | 48 | 31 |
| 2019 | 44 | 69 | 88 | 63 |
| AVG (1990–2010) | 132 ± 90 | 70 ± 27 | 80 ± 15 | 48 ± 14 |

Notes:
[*] Total days the daily maximum soil moisture was less than 10% water at 20 cm depth. Data was available starting in 2006, thus the average is 2006–2016.
[**] Total days the daily maximum temperature was greater than 20 °C.
Data from the Butte 380 USDA SNOTEL station.

of sample date for each elevation to determine temporal differences in beta diversity. Alpha diversity richness was determined for each sample by calculating observed mean richness and standard deviations with sample replicates followed by visual plots of the results as a time series for each sample date (*Berry, 2016*).

Further, canonical correspondence analysis (CCA) was conducted using the weighted UniFrac distance matrices in statistical software Past 4.01 (*Hammer, Harper & Ryan, 2001*). Environmental conditions of daily average soil moisture and soil temperature from the microclimate sensors at each plot and monthly total precipitation data from the SNOTEL database were included in the CCA to understand the influences of soil conditions on community clustering. Due to a lack of microclimate data at the Upper plot, a CCA was not generated for this location. Finally, mean relative abundances were determined by phylum and family bacterial assignments with the 15 most abundant taxa ranked for binned samples (control, spruce, and lodgepole) across date and location using the ampvis2 package (*Andersen et al., 2018*). Differences in mean relative abundances across sampling dates were visually observed, but also statistically confirmed using differential abundance comparisons in R package DESeq2 (*Love, Huber & Anders, 2014*). A $P$ value less than 0.05 was considered significant in all statistical tests.

## RESULTS

### Variation of season, elevation, and accelerated snowmelt

The three-year study encompassed a wide range of weather patterns that impacted local soil temperature and moisture (Fig. 2). Based on the SNOTEL database and microclimate sensor data from each plot, an active monsoon in the summer of 2017 was followed by a dry year with below-average snow water equivalent (SWE) precipitation in 2018 (Table 1). In 2019, SWE rebounded to almost double the prior year. Precipitation averaged 77 ± 25 cm (SWE) over the study period with a higher snow fraction that ranged from 60–70% of total precipitation. SNOTEL soil moisture data were consistent with these trends with above-average dry soil days in 2018.

In addition to climate variability observed in 2017–2019, early snowmelt deployments allowed us to experimentally manipulate the variable of snowmelt timing. Over the three study years, the permeable tarp deployment at the Middle-ES plot accelerated
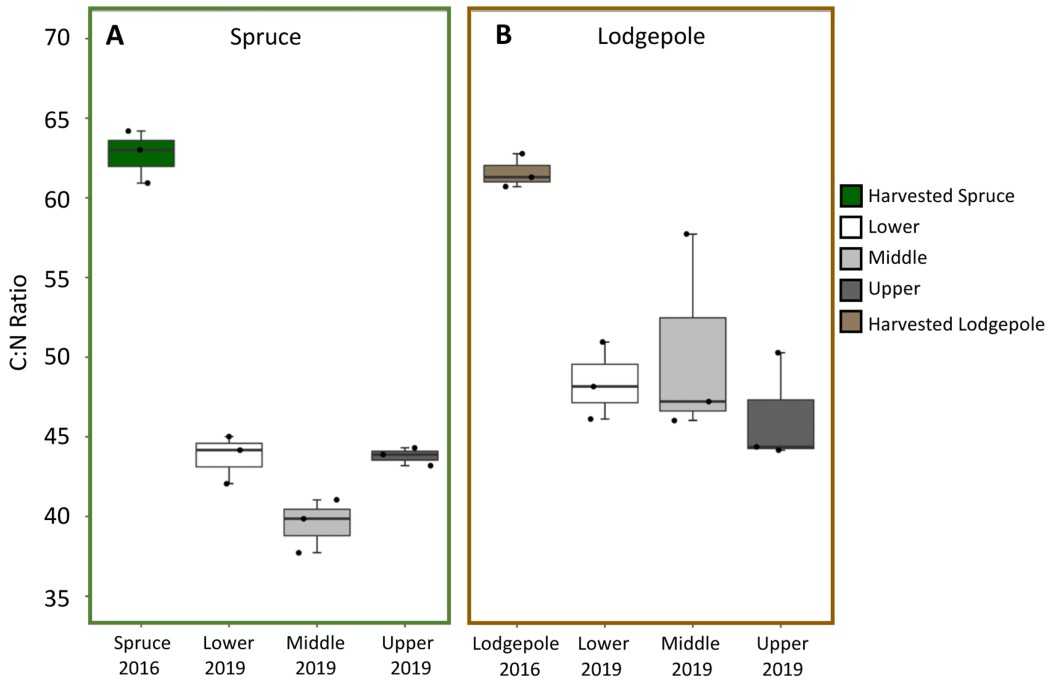

**Figure 3 C:N ratios decreased over the three-year study period across elevations.** Carbon:Nitrogen ratios (C:N) are shown across each elevation for (A) spruce and (B) lodgepole needle decomposition. The harvested 2016 needle values are shown first, followed by the 2019 decomposed Lower, Middle, and Upper plots.

snowmelt by 15 ± 7 days in contrast to the control Middle plot. This was evidenced by the microclimate soil temperature profiles and visual time-lapse capture of plant growth (Fig. S2). Early snowmelt advanced spring plant greening by as much as 22 days in 2017 and both snowmelt and plant growth timing visually mimicked that of the Lower plot (−300 m). The Middle-ES snowmelt plot became indistinguishable from the parallel Middle plot in visual appearance as well as soil moisture and temperature values as the summer progressed.

## Needle decomposition

Spruce and lodgepole needle compositional changes were determined by comparing initial harvested needle carbon and nitrogen totals to the decomposed 2019 samples across experimental variables of elevation and snowmelt. While triplicate deployments limited statistical confirmation, no clear differences in average litter bag mass losses were observed as a function of elevation or snowmelt due to overlapping standard deviations of decomposed samples (Table S2). When needle litter bag mass losses were binned across elevation (Lower, Middle, Upper), this returned a total mass loss after three years of 21% ± 4% for lodgepole and 16% ± 3% for spruce (Table S2). In agreement with mass loss, no pronounced differences in average C:N ratios for lodgepole needles were observed across elevation after decomposition (Fig. 3). This was similarly observed in the spruce needles between the Lower and Upper plots, but interestingly the Middle plot spruce ratio was comparatively lower. No pronounced differences in average C:N ratios were observed

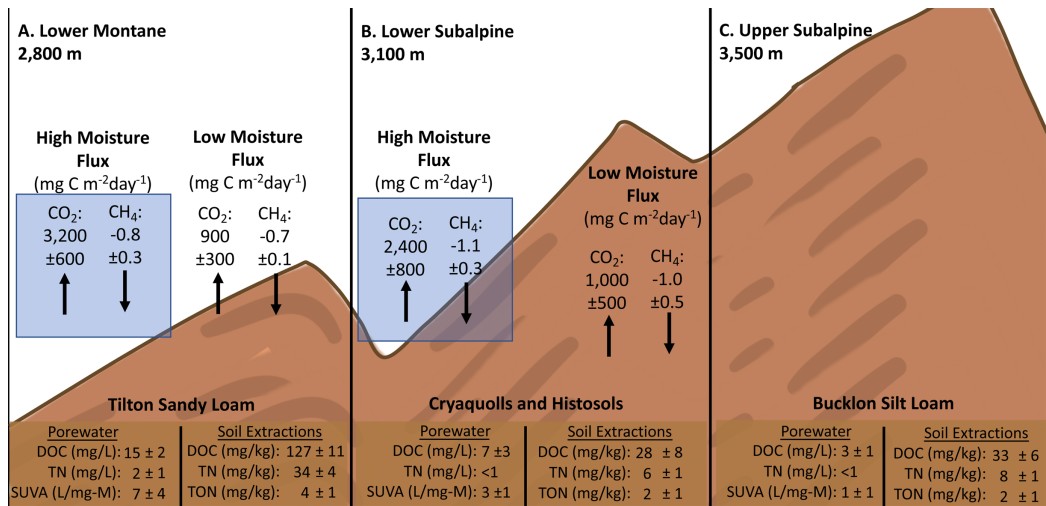

**Figure 4 Compilation of gas flux, porewater, and soil extractions across experimental elevation plots.** Averages of soil porewater and gas flux measurements for each elevation (n=3). Only control values are shown to remove needle decomposition biases.

across snowmelt manipulations. Overall there was a temporal decrease in C:N ratios of approximately 32% ± 4% in spruce and 22% ± 3% in lodgepole after three years of decomposition (Fig. 3). As a result, total carbon decreased in the lodgepole needles by approximately 9% ± 3% and 3% ± 10% for spruce, but with a large variance in spruce from an increase in total carbon at the Upper plot. Total nitrogen increased in both spruce and lodgepole needles by 44% ± 13% and 17% ± 7%, respectively (Table S3).

Needle organic composition was analyzed to further understand decomposition processes. The FTIR spectra for the needle samples after three years of decomposition were similar across elevation and snowmelt manipulations (Fig. S3). Peak integrations were conducted for select organic groups to explore shifts in organic composition over time (Table S4). This revealed a decreased presence of polysaccharides and increased relative presence of amides and aromatics after three years of decomposition. Lodgepole needles contained more labile ether linkages in contrast to a higher presence of recalcitrant lignin-type aromatics in the spruce needles throughout the three years. The ratio of the polysaccharide to amide I peaks served as an estimate of organic C:N ratios. Consistent with total C:N values (Fig. 3), no differences were found across elevation for spruce and lodgepole litter; however, a temporal decrease in this ratio was observed for both sample types (Table S4).

## Porewater & soil extractions

Soil porewater samples were collected to quantify dissolved organic carbon (DOC) and total dissolved nitrogen (TN) concentrations across elevations and study years. Differences in average values were observed across elevation based on standard deviation values with consistently higher concentrations of control sample porewater DOC and TN at the Lower plot (15.1 ± 1.7 mg/L DOC, 2.3 ± 1.2 mg/L TN) in contrast to the Middle (7.2 ± 3.4 mg/L DOC, 0.4 ± 0.3 mg/L TN) and Upper (2.8 ± 1.1 mg/L DOC, 0.3 ± 0.1 mg/L TN) (Fig. 4). As a result, concentrations of DOC and TN decreased with rising elevation.

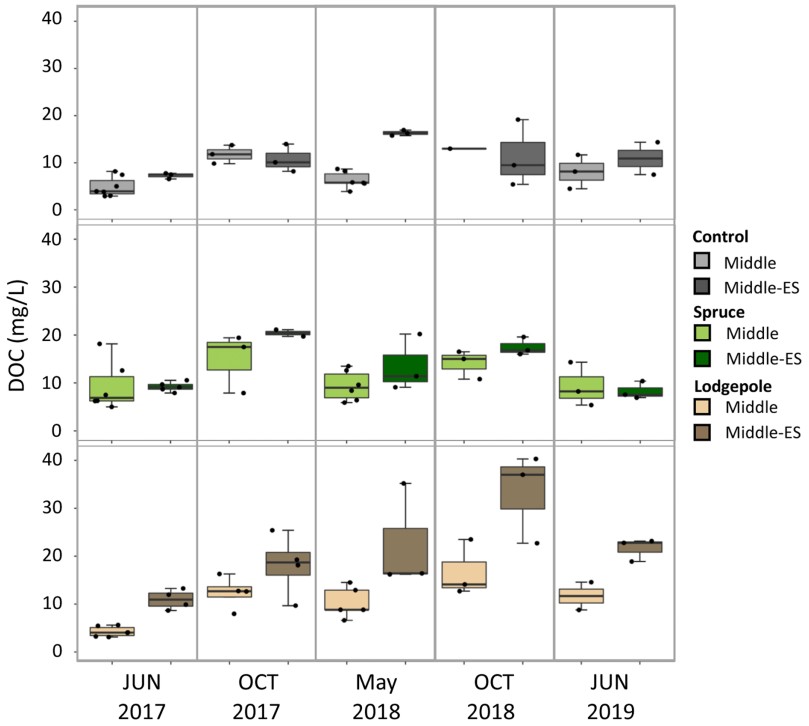

**Figure 5 Higher organic carbon porewater concentrations are associated with the early snowmelt plot in contrast to the control snowmelt plot.** 2017-2019 porewater DOC concentrations were collected for both the Lower Subalpine control (Middle) and early snowmelt (Middle-ES) plots for the control, spruce, and lodgepole samples. Error bars indicate plus or minus one standard deviation. Samples were collected during two consecutive days from both plots in 2017-2018 to increase the sample size (n = 3-6).                                         

No consistent trends were determined as a function of needle presence or type at each study plot. However, there was a suggestive trend with respect to accelerated snowmelt where the Middle-ES plot exhibited higher DOC concentrations than the Middle control plot. This trend was observed across samples with the largest average increases observed under lodgepole needles in the spring sampling events (Fig. 5). While the plots were adjacent but separate, variables beyond snowmelt timing cannot be fully ruled out as a possible explanation. The porewater TN concentrations followed a similar trend but standard deviation values overlapped (Fig. S4).

Analogous to porewater results, sacrificial soil extractions at the end of the experiment exhibited differences across elevation with the highest DOC and TN concentrations at the Lower site. Control soil extractable concentrations were approximately five times higher at the Lower plot when contrasted with the Middle and Upper plots (Table 2). The higher TN concentrations at the Lower site were associated with greater levels of extractable nitrate in contrast to the other plots. Dissolved total organic nitrogen (TON) and nitrate constituted >90% of the nitrogen species within the soil. Accelerated snowmelt did not appear to impact average soil DOC and TN between the Middle and Middle-ES plots (Table 2). In contrast to porewater, the presence of needles resulted in

**Table 2 Soil extractions across elevation and samples.**

| | Lower | | | Middle | | | Middle-ES | | | Upper | | |
|---|---|---|---|---|---|---|---|---|---|---|---|---|
| | Control | Spruce | Lodge | Control | Spruce | Lodge | Control | Spruce | Lodge | Control | Spruce | Lodge |
| DOC (mg/kg) | 127 (±11) | 183 (±39)* | 171 (±18) | 28 (±8) | 68 (±34) | 115 (±41) | 32 (±3) | 54 (±7) | 94 (±11) | 33 (±6) | 56 (±16) | 92 (±21) |
| TN (mg/kg) | 34 (±4) | 33 (±19)* | 24 (±7) | 6 (±1) | 11 (±9) | 7 (±3) | 6 (±1) | 5 (±1) | 4 (±1) | 8 (±1) | 9 (±3) | 16 (±16) |
| TON (mg/kg) | 4 (±1) | 4 (±2)* | 5 (±2) | 2 (±1) | 4 (±3) | 5 (±2) | 2 (±0) | 2 (±1) | 3 (±1) | 2 (±1) | 1 (±2) | 1 (±1) |

Note:
* $n = 2$.
Concentrations of DOC (dissolved organic carbon), TN (total dissolved nitrogen), and TON (total dissolved organic nitrogen) across elevations in August 2020 ($n = 3$).

higher extractable soil DOC across elevations when compared to the needle-free control rings (Table 2). Higher extractable DOC was associated with lodgepole needles relative to spruce.

## Gas flux

To further understand the influence of needle decomposition in conjunction with climate variables on terrestrial carbon cycling, gaseous carbon flux was measured at both the Lower and Middle plots. Gas flux did not differ between the Lower and Middle plots or as a function of snowmelt manipulation based on overlapping standard deviation values (Fig. 6). Peaks in average $CO_2$ production were observed across elevation in association with moisture events which returned flux values 2–3 times higher than rates during low moisture periods (Figs. 2, 4, and 6). The seasonal $CO_2$ flux was further heightened by the presence and type of needle litter (Fig. 6 and Fig. S5). This held true when contrasting the comparatively wet summer of 2017 with drier years in 2018 and 2019 (Fig. 6 and Fig. S6). Pearson correlations returned inconsistent correlations between total annual flux measurements and soil moisture and temperature (Table S5). Observed average $CH_4$ fluxes were not appreciably different as a function of elevation, snowmelt manipulation, needle presence, or type (Fig. S7). Average control $CH_4$ consumption was $-0.7 \pm 0.1$ mg C m$^{-2}$ d$^{-1}$ for the Lower plot and $-1.0 \pm 0.5$ mg C m$^{-2}$ d$^{-1}$ at the Middle plot in 2017–2018 (Fig. 4). Additionally, $NH_3$ and $N_2O$ fluxes exhibited high variability and no clear trends of production or removal.

## Microbial ecological responses

Microbial community clustering was significant across locations as indicated by the Adonis multivariate analysis of variance using weighted and unweighted UniFrac distance matrices (Fig. S8). These differences were observed across elevation for both 16S bacterial and 18S eukaryotic genes ($P < 0.05$). Further comparisons between the two Middle plots revealed that accelerated snowmelt did not significantly impact community clustering. Needle presence and type however influenced community clustering across elevations with a reliance on both date and location. Specifically, when Adonis analyses were conducted for each sample type across each year, significant differences in community beta diversity were observed in 2018 at the Lower and Middle plots. Due to low sample counts, this was not observed at the Upper plot (Table S6).

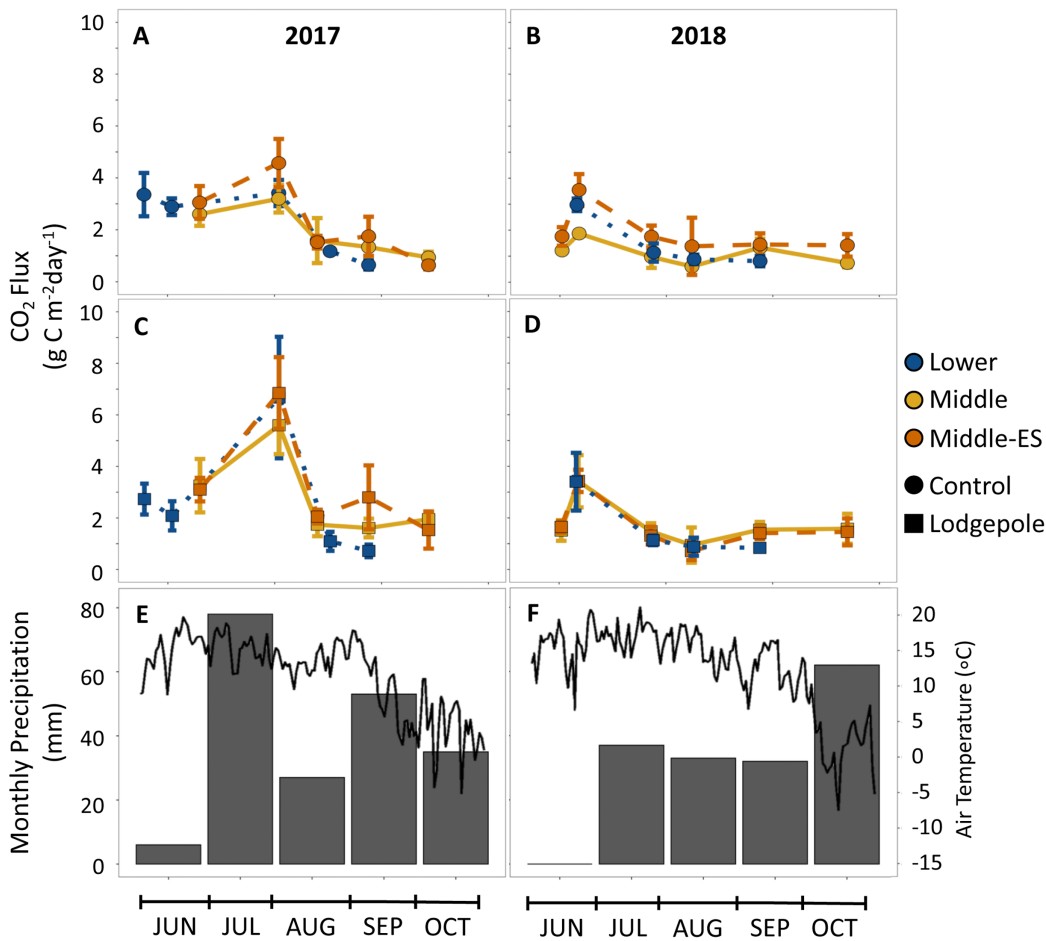

**Figure 6 $CO_2$ production is influenced by seasonal shifts in temperature and moisture, but not elevation.** $CO_2$ flux measurements over the snow-free months of 2017-2018 for the (A, B) control and (C, D) lodgepole samples across elevation and snowmelt manipulations. The 2019 full data and spruce samples are not shown but can be found in Figs. S5 and S6. (E, F) air temperature and total precipitation are shown from the Butte 380 SNOTEL (3,100 m) database. Error bars indicate plus or minus one standard deviation.

The alpha diversity time series (Fig. 7) further highlighted the impact of seasonal shifts on bacterial community structure and diversity in 2018 at the Lower and Middle plots with a relative increase in species richness in July 2018 for all sample types. Analogous trends were observed in the alpha diversity time series at the Upper plots (Fig. S9), but there was high variance in standard deviations. Mean relative familial abundance analyses further detailed community shifts in specific taxa at the Lower and Middle plots by contrasting early-season May 2018 against a drier July and October 2018 (Fig. S10). Upon visual inspection of the heatmaps and statistical confirmation using differential abundance analysis, clear changes occurred in 2018 for specific taxa. As a result, relative abundances of the early-season taxa of primarily *Chitinophagaceae*, among others fell in July or October of 2018 at each elevation with fewer dominant families not observed previously increased in relative abundance. This corresponded to an average increase in community richness that was most pronounced at the Middle elevation (Fig. 7) in 2018 as

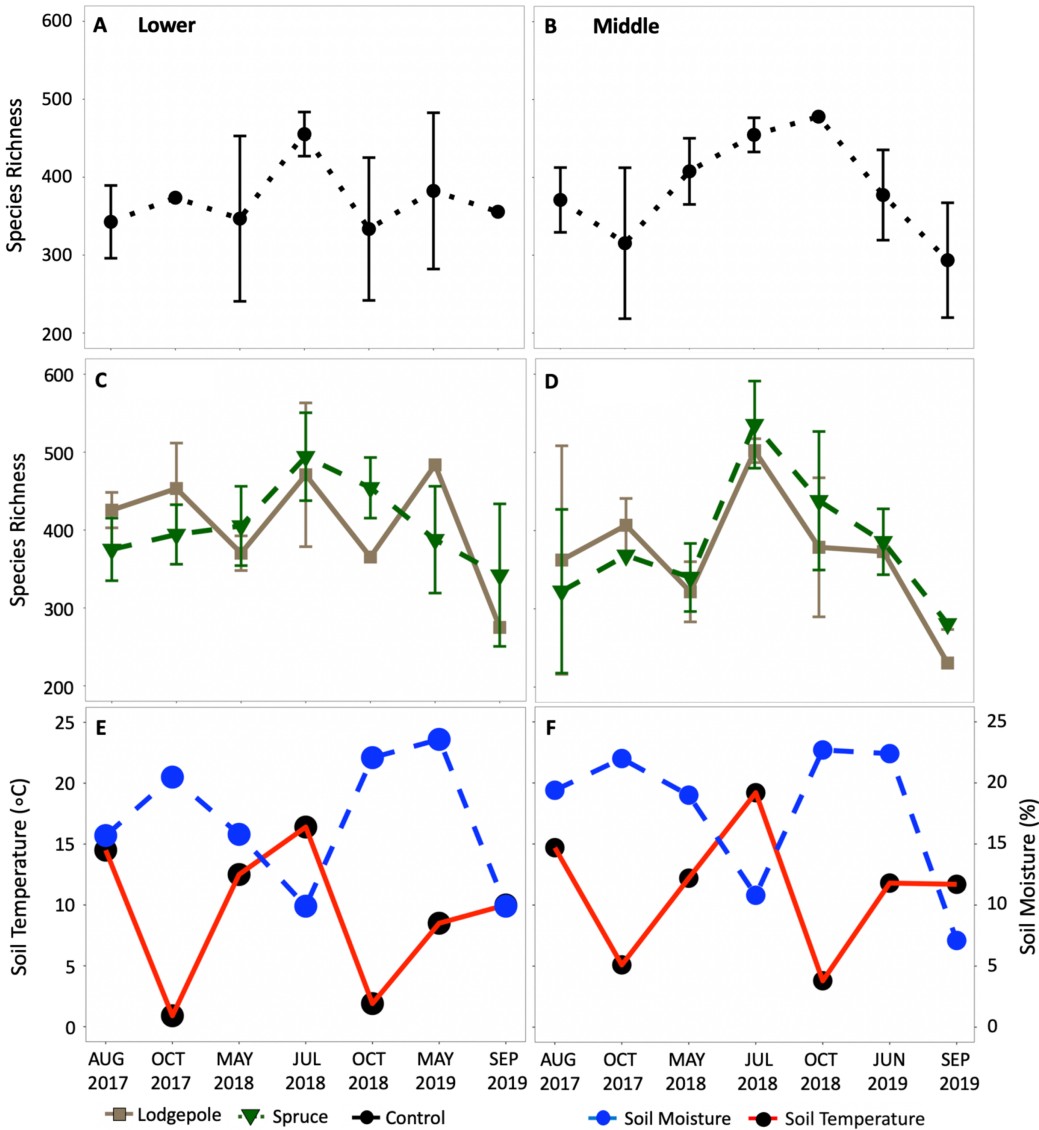

**Figure 7 Average alpha diversity richness increased in the summer dry period of 2018.** Species richness changes over time at the (A, C, E) Lower and (B, D, F) Middle plots for the (A, B) controls and (C, D) needle samples. The average daily soil temperature and moisture are shown in red and blue for the (E) Lower and (F) Middle plots on each sample date. The Upper plot is not shown but can be found in Fig. S9. Error bars indicate plus or minus one standard deviation ($n = 3, 4$). Note a lack of error bars for a sample point is due to $n < 3$ caused by sample removal during rarefaction.

well as significant differences in variance in 2018 at both the Lower and Middle elevations (Table S6). In contrast to the bacterial communities, this trend was less clear in eukaryotic 18S richness (Figs. S9 and S11). Visually, shifts in relative abundances of specific phyla were observed (Fig. S12); however, due to low sample counts during rarefaction, the changes were not statistically confirmed with differential abundance comparisons. Few consistent trends in specific bacteria were found at the genus level except for drought-resistant *Mycobacterium* and *Isosphaera* which were present in multiple samples

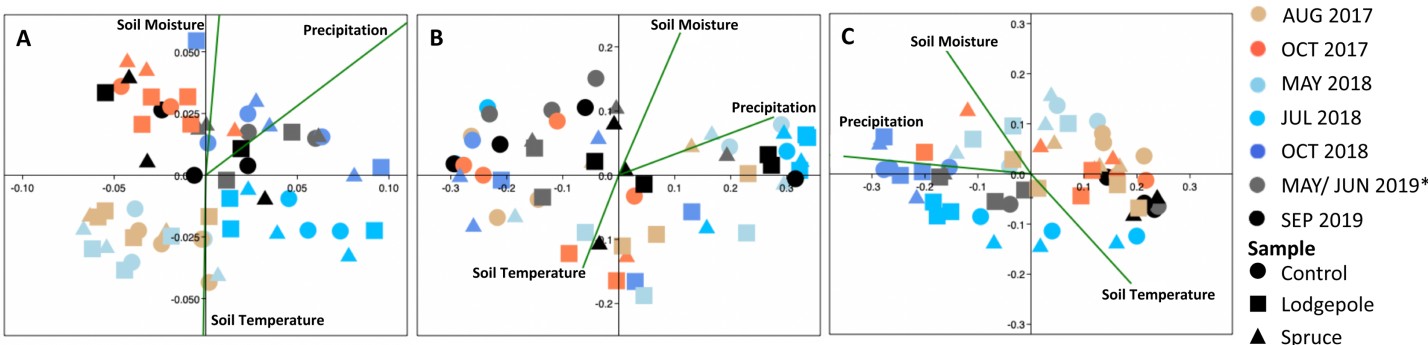

**Figure 8 Bacterial communities cluster significantly as a function of date and environmental variables.** Canonical correspondence analysis of environmental variables drive beta diversity community clustering as a function of soil temperature and moisture over time at the (A) Lower, (B) Middle and (C) Middle-ES plots. Soil moisture and temperature are values unique to each plot from the microclimate sensors on the day of collection. Precipitation represents monthly totals from the Butte 380 SNOTEL (3,100 m) database. *The Lower samples were collected in May 2019, and Middle in June 2019.

at the Middle and Lower plots during the dry phase in July 2018. The observed trends at the Upper plot were less clear due to high variance; however, it is recognized that fewer total samples were collected at this location with incomplete soil moisture and temperature data (Figs. S9 and S11).

The CCA plots highlighted the association of precipitation, soil moisture, and soil temperature with the observed shifts in bacterial community structure in 2018 (Fig. 8). Specifically, at the Lower plot, the 2018 samples clustered by date with May and July 2018 samples associated with lower precipitation (32 ± 5 mm) and soil moisture (14% ± 3%) whereas October 2018 clustered in association with higher precipitation (58 ± 22 mm) and soil moisture (22% ± 1%). Interestingly, the September 2019 microbial samples also clustered in proximity with high soil moisture and sampling dates, despite low precipitation (6 mm) and moisture (10%) conditions. The July 2018 samples at the Middle plot clustered tightly in association with lower precipitation (37 mm) and soil moisture (11%). The Middle-ES plot returned similar associations with the 2018 samples clustered separately from the 2017 and 2019 samples. Due to low sample counts after rarefaction, the 18S CCA plots returned high variability, which limits interpretation (Fig. S13).

## DISCUSSION

This three-year experiment was designed to monitor needle decomposition and associated biogeochemical responses across the climate variables of elevation, snowmelt, and season in simplified soil plots. Our subsequent monitoring window encompassed a wide range of annual variability in snowpack and precipitation that included abnormally dry conditions. As such our results provide insight into carbon, nitrogen, and microbial shifts in montane and subalpine ecoregions during dry soil conditions and early snowmelt that are projected to become more frequent under climate change (*Lukas et al., 2014*). Additionally, by contrasting lodgepole and spruce litter against bare soil, we were able to discern contributions from conifer needle litter to biogeochemical processes.

## The roles of soil type, elevation, and accelerated snowmelt on soil biogeochemistry

The experimental 700-m elevation gradient across montane and subalpine ecoregions harbored clear differences in the baseline soil properties of porewater, extractable organic carbon and nitrogen, and microbial community structure (Fig. 4). While soil characteristics drive bioavailable carbon, nitrogen (*Delgado-Baquerizo et al., 2015*), resident microbial communities (*Thomson et al., 2015*), and soil moisture (*Lawrence & Hornberger, 2007*), trends as a function of elevation are not consistent. Past studies across a 1,000-m gradient determined inconsistent trends in soil organic carbon across elevation (*Djukic et al., 2010*), while a larger gradient of 3,000 m established positive and negative correlations in soil carbon and nitrogen content with altitude and temperature, respectively (*Tashi et al., 2016*). In our present study, the elevation gradient of 300 m between the Lower and Middle plots did not appreciably impact gas flux (Fig. 6), while mass losses from deployed conifer needles were similar across the full 700-meter gradient (Table S2). This was surprising considering the differences in snowmelt timing across elevation (Table S1); however, these results agree with other studies that have shown minimal impact on microbial structure with no lasting effects on microbial biomass and nutrient cycling processes after experimental snowmelt manipulations (*Conner, Gill & Harvey, 2017*; *Darrouzet-Nardi et al., 2019*). A plausible explanation is that ecological resilience overcomes the effects of accelerated snowmelt without additional climate variables of higher temperatures and lower precipitation (*Darrouzet-Nardi et al., 2019*).

Snowmelt timing has been shown to influence litter decomposition rates associated with freeze-thaw cycles by releasing more nutrients to bacteria that become active during thawing (*Zhu, Yang & He, 2013*). While the decomposition rates over three years of our study were not considerably different between the Middle plots, soil porewater DOC consistently trended higher under accelerated snowmelt in the spring. These trends continued in the fall, but with higher variability in significance (Fig. 5). While the Middle-ES plot could have locational bias, the two plots (Middle and Middle-ES) were in proximity, of the same soil type, and the greatest differences were observed only under the lodgepole needles. Carbon losses can be enhanced with spring timing, early-season frost events, and late-season drought (*Arnold, Ghezzehei & Berhe, 2014*), all of which are location-dependent. Collectively, these observations could be explained by a climate-related phenomenon associated with more labile litter decomposition. Conifer litter decomposition depends on the quality, or bioavailability of organic matter and nutrients, resulting in different biogeochemical responses as a function of tree needle type (*Leonard et al., 2020a*). As a result, lower quality needles can limit bioavailable organics, while soil type is more influential in potential decomposer limitations under higher quality litter (*Vesterdal, 1999*). Collectively, this suggests that accelerated snowmelt could increase access to nutrients under high quality litter during the spring thaw cycle for microbes becoming more active under early snowmelt. Interestingly, this is in contrast to the findings of *Conner, Gill & Harvey (2017)* in which no changes were observed in soil DOC over three years of early snowmelt in a subalpine aspen forest. These conflicting

results reveal the limitations in this study from the exclusion of native foliage, as the observed effects of early snowmelt on carbon cycling can be mediated by plant growth (*Wipf, Rixen & Mulder, 2006*; *Conner, Gill & Harvey, 2017*).

## The importance of needle litter and seasonality

While differences in soil temperature and moisture across a 700-meter gradient exerted little influence, needle presence and type increased soil DOC and $CO_2$ fluxes across the elevation transect. Despite overlapping errorbars, there was a trend toward higher mass losses and higher rates of heterotrophic respiration in association with lodgepole needles across elevations (Fig. 6). This agrees with other studies at lower forest elevations (*Fujii et al., 2018*), subarctic tundra (*Sundqvist, Giesler & Wardle, 2011*), and a subalpine elevation gradient (*Zhu, Yang & He, 2013*) that collectively concluded that plant litter type affects decomposition rates more than abiotic influences found across climate regimes. In support of higher mass loss rates, needle decomposition also influenced heterotrophic respiration with as much as twice the $CO_2$ production under lodgepole samples in contrast to needle-free controls. This is consistent with earlier findings confined to the Middle plot (*Leonard et al., 2020a*) where lodgepole needles provided a more labile carbon source than spruce. Further, as expected from previous findings, peaks in $CO_2$ production were associated with wetter soils from snowmelt and precipitation events (Fig. 6) while $CH_4$ consumption remained relatively consistent. This supports prior findings where $CO_2$ production is more dependent on climate whereas $CH_4$ consumption relies more strongly on soil characteristics (*Gulledge & Schimel, 2000*).

Heterotrophic respiration depends on both water availability (*Liu, Zhang & Wan, 2009*) and temperature (*Schindlbacher et al., 2011*). As a result, both variables likely influence respiration rates as driven by soil saturation thresholds. This was observed in a prior boreal forest study where $CO_2$ production in dry soil conditions, defined as below soil half-saturation, was more sensitive to moisture, while wetter conditions were more dependent on temperature (*Gulledge & Schimel, 2000*). This phenomenon could explain our observation of conflicting correlations across plots. Alternatively, or in tandem, this could be explained by interchanges between abiotic and microbial activity-related influences on $CO_2$ rates where high moisture release is driven more by microbial respiration and low moisture release by abiotic mechanisms. This is consistent with the observed slowing of microbial activity during dry periods in previous studies (*Davidson, Belk & Boone, 1998*; *Wu et al., 2011*; *Carbone et al., 2011*). As brought to question by *Schimel (2018)* the mechanisms which increase $CO_2$ during rewetting events are unclear. However, our study suggests increased biotic mechanisms under needle decomposition by demonstrating that areas with more labile litter are associated with higher carbon fluxes during moisture events. It has been projected that seasonal shifts in soil moisture will increase overall soil respiration in montane and subalpine ecoregions such as those analyzed here (*Kramer & Chadwick, 2018*). In our study, dry soil conditions were associated with low baseline fluxes and repressed heterotrophic activity. This could become more frequent with projected increases in temperature if precipitation remains the same or decreases. By extension, there could be decoupling from temperature where moisture

would be the main determinant of microbial activity and $CO_2$ respiration rates (*Gulledge & Schimel, 2000*). While drought could repress carbon release, this could be associated with more pronounced export during rewetting (*Meisner et al., 2017*).

## Microbial resilience and response to drought

A temporal analysis of microbial community structure across deployments provided further insights. Consistent with heterotrophic respiration rates, resident bacterial communities were influenced by needle presence, type, and seasonal shifts in soil moisture. This had previously been explored at the Middle plot in *Leonard et al. (2020a)*. Our present study furthered insights across elevation with significant community variability depending on the sample date (Table S6). The CCA plots (Fig. 8), and alpha diversity time series (Fig. 7) emphasized the importance of reduced moisture and higher temperatures in bacterial ecology, which is supported by experimental simulations of drought (*Von Rein et al., 2016*) and a field study (*Acosta-Martínez et al., 2014*). Clear differences in bacterial communities were observed after the drier conditions of July 2018 as evidenced by significant beta diversity variance and increased alpha diversity species richness (Table S6, Fig. 7). There are inconsistencies in the literature where drought has been documented to both increase and decrease overall microbial richness (*Peñuelas et al., 2012*; *Ochoa-Hueso et al., 2018*). Specifically, at the Middle and Lower plots, drought-resistant *Mycobacterium* of Actinobacteria (*Torvinen et al., 2006*) and *Isosphaera* of Planctomycetes (*Dai et al., 2019*) ranked as the top 15 most abundant taxa only in July 2018. Interestingly, during this period the bacterial family *Chitinophagaceae* experienced its most pronounced decrease. Analogous findings were observed for fungal communities where the relative abundance of Ascomycota *Dothideomycetes* and Basidiomycota *Tremellomycetes* increased in July 2018 and maintained higher abundances through 2019 at the Middle and Upper elevations under the needle samples. This supports prior field drought-related findings of *Acosta-Martínez et al. (2014)*.

More broadly, at the Phylum level, the July 2018 shift in community structure translated to a decreased abundance of Bacteroides across elevation and samples; agreeing with observed decreases in Bacteroides under elevated temperatures in other studies (*Oliverio, Bradford & Fierer, 2017*). Interestingly, this contrasts with a drought cycle study on agricultural land that reported increased Bacteroides (*Acosta-Martínez et al., 2014*). That same study however highlighted Acidobacteria as sensitive to temperature changes which was consistent with observed increases in our system in 2018. These conflicting results highlight the importance of soil type and location as well as limitations in interpreting and predicting taxonomic shifts associated with a changing climate where heterogeneous mountain terrain is an example. Despite the changes associated with July 2018, a return of the previously dominant pre-drought taxa and abundances in 2019 provided evidence for microbial resilience across our elevation gradient. This agrees with previous findings in which diverse communities are more resilient partially due to rare taxa that can continue functional processes and increase in abundance in response to disturbances (*Shade et al., 2012*). As a result, microbial communities have an inherent resilience to disturbances (*Allison & Martiny, 2008*) and greater tolerance than local flora and fauna

(*Schimel, 2018*). However, the dynamics of plant and microbial interactions, though not explored in our current study, could impact these findings (*De Vries et al., 2018*).

## CONCLUSIONS

This study highlights the sensitivity, but also the resilience of biogeochemical decomposition processes to elevation, season, and snowmelt. An elevation gradient of 700 m or less will likely have a minimal effect on the biogeochemical responses to drought conditions. Earlier snowmelt in the timescale of weeks is not expected to significantly affect local biogeochemistry beyond an increased potential for DOC release in organic horizons with more labile litter influences, whereas climate plays a more prominent role in soil baseline conditions. While drier soil conditions reduced heterotrophic respiration, carbon release could be magnified when moisture rewetting returns. This is especially significant in high sensitivity biomes that include the studied montane and subalpine ecoregions in Colorado. As a result, with higher dependency on moisture events, increased carbon release in tighter timeframes could occur under litter decomposition. The microbial communities are likely resilient to drought during short-term disturbances if soil saturation returns, but the impacts of long-term drought need to be further understood. If increased drought stress from climate change continues in Colorado, this could reinforce the lower respiration rates and microbial community shifts observed in the 2018 dry season of this study.

## ACKNOWLEDGEMENTS

Field access and support were provided by the Rocky Mountain Biological Laboratory in Gothic, CO with logistical support from Jennifer Reithel. We thank Kristin Mikkelson, Brent Brouillard, Chelsea Wilmer, Heidi Steltzer, Kayla Hubbard, Jake Wands, and Sabrina Nesladek for their support of field activities. Zhao Hao conducted FTIR measurements and provided analysis techniques. Comments from Gary Vanzin and Zackary Jones on a prior draft improved data analysis and dissemination. Members of the Geo-Environmental Microbiology Laboratory at the Colorado School of Mines provided ongoing support and friendship. We thank PeerJ editor Mark Tibbett and two anonymous reviewers for their constructive suggestions during the peer review process.

### Funding

Financial support was provided by the U.S. Department of Energy (DOE), Office of Science, Office of Biological and Environmental Research under exploratory university-led research: DE-SC0016451 with support through the Lawrence Berkeley National Laboratory's Watershed Function Scientific Focus Area under contract DE-AC02-05CH11231 (Lawrence Berkeley National Laboratory; operated by the University of California). The funders had no role in study design, data collection and analysis, decision to publish, or preparation of the manuscript.

## Grant Disclosures

The following grant information was disclosed by the authors:
U.S. Department of Energy (DOE), Office of Science, Office of Biological and Environmental Research: DE-SC0016451.
Lawrence Berkeley National Laboratory's Watershed Function Scientific Focus Area: DE-AC02-05CH11231.

## Competing Interests

Eoin L. Brodie and Jonathan O. Sharp are academic editors for PeerJ.

## Author Contributions

- Laura T. Leonard conceived and designed the experiments, performed the experiments, analyzed the data, prepared figures and/or tables, authored or reviewed drafts of the paper, and approved the final draft.
- Eoin L. Brodie conceived and designed the experiments, authored or reviewed drafts of the paper, and approved the final draft.
- Kenneth H. Williams conceived and designed the experiments, authored or reviewed drafts of the paper, and approved the final draft.
- Jonathan O. Sharp conceived and designed the experiments, prepared figures and/or tables, authored or reviewed drafts of the paper, and approved the final draft.

## Field Study Permissions

The following information was supplied relating to field study approvals (*i.e.*, approving body and any reference numbers):

The Rocky Mountain Biological Laboratory and the U.S. Department of Agriculture and Forest Service (Authorization ID: GUN1120) approved the fieldwork for this study.

## Data Availability

The DNA extraction sequence data is available at NCBI SRA: PRJNA605259 and PRJNA715914.

The additional raw geochemical and seasonal datasets are available in the Supplementary Files.

The written codes used to analyze the DNA sequence and climate data are available at GitHub: https://github.com/ltleonard/Leonard-et-al.-Climate

## Supplemental Information

Supplemental information for this article can be found online at http://dx.doi.org/10.7717/peerj.11926#supplemental-information.

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
