# Peer review of "Effect of elevation, season and accelerated snowmelt on biogeochemical processes during isolated conifer needle litter decomposition"

_PeerJ, doi:10.7717/peerj.11926_

## Round 0.1 · original submission · Major Revisions

Your manuscript is of interest but requires major revision. The methodology is not always clear to follow and results are not supported by statistical analysis. These points need addressing along with other comments from your reviewers.

Reviewer 1 ·

Basic reporting

The manuscript reports on the influence of elevation, soil parameters (moisture, temperature), snowmelt timing, precipitations on coniferous leaf litter decomposition and soil microbial communities. As the authors note, in general, relevant (modelling) studies address soil C losses ignoring local environmental heterogeneity. Thus, this work has the potential to contribute to filling this knowledge gap.
The results of no elevation and snow-melting time effects on leaf litter decomposition are novel and surprising.
Unfortunately,
(1) the number of replicates was insufficient to carry out a consistent statistical assessment (in some measurements);
(2) FTIR analysis is not specific enough. For example, there are numerous peaks assigned to polysaccharides, and several of them need to be integrated to obtain a reliable value. How relevant is the peak at 1600 cm for amides when there are the NH2 stretches/scissors at 3370-3170/1620-1650, C=O at 1680-1630, C-N at 1430-1390 cm-1and several peaks for amide IV and V at 600-900? Commonly, a peak ratio is more informative (see an IR spectral decomposition index Margenot et al. 2015 SSSAJ).
(3) the eukaryotic (decomposing fungal) communities are not shown;
(4) a link between soil microbial communities and decomposition is missing. We have a microbial ecological response and an indirect interpretation through environmental effects. Still, no gas flux or other measurements were included (I’ve seen that you concomitantly collected samples for DNA and gas flux measurements on a few occasions).
The manuscript is clear, professional English language used throughout.
Introduction
- The context could be better explained. How does snow melting affect the local environment? Which is the meaning of “high moisture”. Commonly, at high moisture content, microbial activity decreases because of low oxygen supply. I am not aware of any dramatic soil moisture effects on bacterial diversity or community composition, but their activity.
- Provide the reasons for which litter type influence microbial decomposition processes.
- At this stage, we do not know whether microbial communities are resilient or not to low moisture conditions, or do we know it from literature?
- Line 73-75: This phrase is exciting. I read it a few times to understand it. Is it really necessary?

Experimental design

The investigation is novel and rigorously conducted. The authors acknowledged the limitation of the design that resulted in a low number of replicates.
- Line 117: how large was the mesh size? Did soil fauna have access to the leaf litter?
- L 171: which R function did you use?
I am not aware of the raw data supply.

Figures
- The captions for supplementary figures are missing.
- Fig. 2: please indicate that the arrows represent the sampling events.
- Fig. 3: C:N ratio is essential, but discussing litter decomposition, changes in C and N are also relevant (maybe you can show it in the supplement, and discuss the N enrichment together with the observations in the FTIR amide peaks – just a suggestion).

Validity of the findings

No comment.

Additional comments

see above

Reviewer 2 ·

Basic reporting

The manuscript is written in clear English language. Introduction part is well elaborated with relevant literature referenced and with clearly stated aims and hypothesis. Figures in main text are relevant and well described. Besides figures in main text authors provided many figures and tables in supplementary information which I appreciate, however the description to supplementary material is missing, which makes it difficult to evaluate presented results and data. Access to raw data is provided.

Experimental design

The manuscript deals with important and interesting topic that fits into the scope of journal. I appreciate the diversity and amount of data the authors collected in the sites that are difficult to access. However, material and method section contains some unclear steps and some of the methods are not described with sufficient detail- see below the specific comments.

line 89-90 Why was the litter collected in different location than the study site?

line 99 unclear what 25x18cm ring mean? What was the diameter of the ring?

line 102-103 unclear the calculation of 48 rings – you write you had 3 elevations, 4 replicates, 3 treatments (spruce, lodgepole, control) which gives together 36

line 109-110 Were the plants removed including the roots? If so this can affect soil microbial communities and respiration rates as rhizosphere microbiomes represent hotspots of microbial activity in the soil. If only above ground parts of plants were removed this can still affect the soil microbes as some of them may use photosythates as a source of nutrient. I find the removal of plants as very artificial treatment. Can you explain the reason for removing the plants?

line 111-113 can you be more specific about the dates of individual measurements and soil extractions?

line 118-119 what was the size of the mesh?

line 212-215 using such approach for soil sampling, how do you know that the collected volume of soil from each plot was the same, thus the results of microbial communities are comparable? Was the soil mixed and homogenized?

line 215 how many DNA extractions were done from each sample?

line 219 how many PCR reactions were done from each DNA sample?

Validity of the findings

The biggest problem I have with the manuscript is how the data was analyzed and presented. In several cases authors claim results that are not supported by statistical analysis or the used statistical analysis is not described- find below the specific examples. In general, I feel the manuscript lack the connection of collected datasets and the use of relevant statistical analysis that would allow to draw the conclusions.

line 280 statement: No clear differences.... Does that mean there were no significant differences? How was the significance for differences in needle decomposition calculated? I am unable to find it in material and methods.

line 283-285 Based on what analysis you concluded there were no pronounced differences in C/N needle ratios? In methods you write that possible significance between samples is indicated by non-overlapping error bars of box plots and in Figure 3 the error bars of spruce middle x lower and middle x upper do not overlap. Your statement of no differences is confusing, please explain.

line 300-301 differences were clear across elevation... based on what analysis you concluded this?

line 306-309 I presume that the significance is claimed based on box plot figures.... if so I feel such conclusion is a bit exaggerated and you should rather write possible significance.

line 325 and 332 unclear how was the significance assessed

line 339-342 how was this significance / non significance analyzed? I miss this information in material and methods.

line 344-346 In the text you mention the results of principal coordinate analyses but you refer to figure and table showing something else.

line 346-347 again not clear how was the significance tested

line 350-351 if you claim decrease or increase of microbial taxa you need to support it by statistical analysis, from the data I see in Figure S9 which represent I presume means of relative abundances you can not make such conclusion.

line 357 how was the significance calculated?

---

## Round 0.2 · Minor Revisions

As you will see from the reviewers comments there are still some aspect that needs to be revised. Currently the description of the supplementary material makes it difficult to evaluate the results and data.

Reviewer 1 ·

Basic reporting

No Comment

Experimental design

No Comment

Validity of the findings

No Comment

Additional comments

The authors addressed all my critical points and specific comments.
The data suffer from a low number of replicates. Still, given that the study is under field conditions and the authors made all the efforts for sound statistical analysis, this study has its merit. The statistical analysis is explained in detail, and the scientific message is clear.
In my opinion, the manuscript is acceptable in its present form.

Reviewer 2 ·

Basic reporting

no comment

Experimental design

no comment

Validity of the findings

no comment

Additional comments

In the revised version of manuscripts authors replied to reviewers comments and addressed most of the raised issues however there are still several things that needs to be revised. I still can not find the description of supplementary material, which makes it difficult to evaluate presented results and data. Find below the specific comments - line numbers correspond to the manuscript with tracked changes.

line 58.... add reference
line 61 ... delete dot before the reference
line 279-281... can you be more specific which particular analysis were conducted using the listed R packages? I am unable to find any particular results from DESeq2 analysis in the manuscript
line 310-311.... what data for soil moisture and soil temperature were exactly used? Was it average of some period of time?
line 383.... what is visual difference?
line 461 – 465... neither in figure S10 nor anywhere else in the text I am unable to find any results from differential abundance analysis that would show statistical significant differences in abundance of microbial taxa
line 465-466.... based on what analysis did you conclude that there was an overall increase in community richness in 2018? – I can not find any analysis that would support this conclusion

---

## Round 0.3 · accepted · Accept

Thank you for dealing so thoroughly with the second set of comments. Your paper will an important contribution to understanding the complexities of litter decomposition.